# Genome-wide association study identifies risk loci for progressive chronic lymphocytic leukemia

Wei-Yu Lin [1], Sarah E. Fordham[1], Nicola Sunter[1], Claire Elstob [1], Thahira Rahman[1], Elaine Willmore[1], Colin Shepherd[1], Gordon Strathdee [1], Tryfonia Mainou-Fowler[1], Rachel Piddock[1], Hannah Mearns [1], Timothy Barrow[2], Richard S. Houlston [3], Helen Marr[4], Jonathan Wallis[4], Geoffrey Summerfield[5], Scott Marshall [6], Andrew Pettitt[7], Christopher Pepper [8], Christopher Fegan[9], Francesco Forconi [10], Martin J. S. Dyer [11], Sandrine Jayne[11], April Sellors[11], Anna Schuh [12], Pauline Robbe[12], David Oscier[13], James Bailey [14], Syed Rais[14], Alison Bentley[15], Lynn Cawkwell[16], Paul Evans[17], Peter Hillmen[18], Guy Pratt[19], David J. Allsup [14,15,20✉] & James M. Allan [1,20✉]

Prognostication in patients with chronic lymphocytic leukemia (CLL) is challenging due to heterogeneity in clinical course. We hypothesize that constitutional genetic variation affects disease progression and could aid prognostication. Pooling data from seven studies incorporating 842 cases identifies two genomic locations associated with time from diagnosis to treatment, including 10q26.13 (rs736456, hazard ratio (HR) = 1.78, 95% confidence interval (CI) = 1.47–2.15; $P = 2.71 \times 10^{-9}$) and 6p (rs3778076, HR = 1.99, 95% CI = 1.55–2.55; $P = 5.08 \times 10^{-8}$), which are particularly powerful prognostic markers in patients with early stage CLL otherwise characterized by low-risk features. Expression quantitative trait loci analysis identifies putative functional genes implicated in modulating B-cell receptor or innate immune responses, key pathways in CLL pathogenesis. In this work we identify rs736456 and rs3778076 as prognostic in CLL, demonstrating that disease progression is determined by constitutional genetic variation as well as known somatic drivers.

[1] Translational and Clinical Research Institute, Newcastle University Centre for Cancer, Faculty of Medical Sciences, Newcastle University, Newcastle upon Tyne, UK. [2] Faculty of Health Sciences and Wellbeing, University of Sunderland, Sunderland, UK. [3] Division of Genetics and Epidemiology, The Institute of Cancer Research, London, UK. [4] Department of Haematology, Freeman Hospital, Newcastle upon Tyne, UK. [5] Queen Elizabeth Hospital, Gateshead, UK. [6] City Hospitals Sunderland NHS Trust, Sunderland, UK. [7] University of Liverpool, Liverpool, UK. [8] Brighton and Sussex Medical School, University of Sussex, Brighton, UK. [9] Institute of Cancer and Genetics, School of Medicine, Cardiff University, Cardiff, UK. [10] Cancer Sciences Unit, Cancer Research UK and NIHR Experimental Cancer Medicine Centres, University of Southampton, Southampton, UK. [11] The Ernest and Helen Scott Haematological Research Institute, Leicester Cancer Research Centre, University of Leicester, Leicester, UK. [12] University of Oxford, Oxford, UK. [13] Royal Bournemouth Hospital, Bournemouth, UK. [14] Hull University Teaching Hospital NHS Trust, Hull, UK. [15] Centre for Atherothrombosis and Metabolic Disease, Hull York Medical School, Hull, UK. [16] University of Hull, Hull, UK. [17] Haematological Malignancy Diagnostic Service Laboratory, St James' Institute of Oncology, Leeds, UK. [18] Section of Experimental Haematology, Leeds Institute of Medical Research at St James's, University of Leeds, Leeds, UK. [19] University of Birmingham, Birmingham, UK. [20]These authors contributed equally: David J. Allsup, James M. Allan. ✉email: hycda1@hyms.ac.uk; james.allan@newcastle.ac.uk

C hronic lymphocytic leukemia (CLL) is a prevalent hematological malignancy with approximately 3750 new cases diagnosed in the United Kingdom every year[1]. CLL is characterized by marked clinical and biological heterogeneity with a proportion of patients having stable, asymptomatic disease which never requires therapeutic intervention, whilst others suffer from a progressive disease with associated shortened survival. Treatment for symptomatic CLL has been transformed by the advent of highly effective chemoimmunotherapy regimens, B-cell receptor signaling pathway inhibitors (BCRi) and agents targeting anti-apoptotic proteins but the majority of patients who progress to the point of therapeutic intervention have historically experienced CLL-related mortality. However the advent of highly effective combination regimens utilizing both BCRi and anti-apoptotic agents may improve patient outcomes[2].

Prognostication in early stage CLL and identifying those patients at risk of progression remains challenging. Numerous methods of CLL prognostication exist, which range from clinical assessments to genomic techniques. Clinical assessments include the Rai[3] or Binet[4] scores, where adverse outcomes are associated with increasing tumor load and evidence of marrow failure, and the CLL international prognostic index (CLL-IPI)[5] which utilizes a combination of clinical and biological characteristics to assess prognosis[6–9]. Cases with unmutated immunoglobulin heavy chain variable region (IGHV) genes[10] and those with mutations/deletions affecting TP53, NOTCH1, and SF3B1 have poor prognoses[11,12], although these alterations are infrequent in newly diagnosed CLL patients[13,14]. Once patients progress to symptomatic disease then somatic drivers of disease, particularly mutations of TP53 and SF3B1, as well as complex karyotype, become dominant prognostic markers[15,16].

CLL arises from an unidentified cell-of-origin in response to continued signaling through the BCR either in response to ongoing antigenic stimulation by auto-antigens or allo-antigens, or is driven by autonomous BCR activation[17]. The increased incidence of CLL in first-degree relatives of affected patients also points to an element of genetic susceptibility, which has been borne out in large scale genome-wide association studies (GWAS) which to date have identified over 40 risk alleles[18–25]. Given the important genetic contribution to CLL susceptibility we hypothesized that constitutional genetic variants also contribute to determining risk of disease progression.

In this work, we conduct a GWAS using data from a large United Kingdom multicentre cohort study of well characterized predominantly early-stage CLL cases to identify two risk alleles for progressive disease in patients with previously untreated disease.

## Results

**Meta-analysis of CLL genome-wide association studies.** We conducted six genome-wide association studies for single nucleotide polymorphisms (SNPs) associating with progressive CLL incorporating cases of European ancestry diagnosed at clinical centers across the United Kingdom (Supplementary Fig. 6; Supplementary Table 1). Of these, data on time to first treatment (TTFT) was available for 755 cases. We combined the association test statistic for 5199911 autosomal SNPs common to all 6 GWAS after exclusion of those with an imputation info (imputation quality) score of <0.9 and a minor allele frequency (MAF) of <0.025, and conducted a meta-analysis under a fixed-effects model. Quantile-quantile plots of observed vs. expected P values for SNPs showed minimal inflation of the test statistic ($\lambda_{GC} = 1.027$) excluding the possibility of hidden population substructure or cryptic relatedness (Supplementary Fig. 9).

Pooling data from 6 GWAS identified 5 SNPs at two genomic locations that surpassed genome-wide significance ($P \leq 5 \times 10^{-8}$)

for association with time from diagnosis to first treatment (Fig. 1). The strongest statistical evidence for an association with progressive CLL was for rs736456 (hazard ratio (HR) = 1.76, 95% confidence interval (CI) = 1.45–2.14; $P = 1.26 \times 10^{-8}$), which maps to the TACC2 locus on chromosome 10q26.13 (Fig. 2). The second strongest association with progressive disease was for rs3778076 (HR = 2.03, 95% CI = 1.58–2.62; $P = 3.89 \times 10^{-8}$) which maps to the SPDEF and C6ORF106 (ILRUN) genes on chromosome 6p (Fig. 2). We genotyped rs736456 and rs3778076 in a seventh cohort, bringing the total number of CLL cases analyzed to 842. Both markers showed consistent direction and magnitude of effect sizes across all seven studies with no evidence of heterogeneity (Fig. 3). There was no evidence of significant interaction between rs736456 and rs3778076 ($P = 0.131$), suggesting that each locus has an independent effect on CLL progression. Analysis conditioning on the lead SNP at each locus showed no evidence for other variants ($P < 10^{-4}$) associating with progressive disease (Supplementary Fig. 10).

Neither the lead SNP on chromosome 10 nor chromosome 6 were significantly associated with post-treatment survival (rs736456 (chr. 10), HR 1.03, 95% CI 0.79–1.35, $P = 0.801$; rs3778076 (chr. 6), HR 0.99, 95% CI 0.70–1.42) (Supplementary Fig. 11).

**rs736456 and rs3778076 predict disease progression in early stage CLL.** In order to further investigate the associations with progressive disease we stratified patients by disease stage at presentation and established prognostic markers. rs736456 (chromosome 10) and rs3778076 (chromosome 6) significantly associate with disease progression in patients with Binet stage A disease at presentation, but have limited prognostic impact in patients with Binet stage B and C at presentation (Supplementary Fig. 12) ($P = 3.81 \times 10^{-5}$ in Binet A and $P = 0.12$ in Binet B/C for rs736456 carriers vs. non-carriers; $P = 1.93 \times 10^{-9}$ in Binet A and $P = 0.69$ in Binet B/C for rs3778076 carriers vs. non-carriers), consistent with a specific role in driving progression in early stage disease. Both genetic markers retained significance in multivariate models for disease progression restricted to Binet stage A patients and when adjusted for study in the model (Fig. 4). rs736456 and rs3778076 were particularly powerful markers for disease progression in stage A patients when considered together, where carriers of 1 risk allele ($P = 1.4 \times 10^{-7}$) or 2 or more risk alleles ($P = 7.9 \times 10^{-8}$) have a significantly shorter time to first treatment compared to non-carriers (Fig. 4c). Whilst not as powerful as IGHV status, rs736456 and rs3778076 had prognostic utility approximately equivalent to CD38 status. Moreover, rs736456 and rs3778076 both significantly associate with progressive disease in patients with otherwise low risk clinical markers, including IGHV mutation (Supplementary Fig. 13), CD38 negativity (Supplementary Fig. 14), low β2 microblobulin (Supplementary Fig. 15) and wild-type for TP53 (Supplementary Fig. 16). As such, rs736456 and rs3778076 are particularly powerful prognostic markers in analysis restricted to patients with disease that is Binet stage A, IGHV mutated and CD38 negative (Supplementary Fig. 17). Taken together, these data identify rs736456 and rs3778076 as prognostic for disease progression in patients with otherwise low risk markers, but which have limited prognostic power in patients with high-risk markers or late stage disease.

**Progressive disease loci at chromosomes 10 and 6.** To identify cis-regulated genes at each locus associated with progressive disease we interrogated gene expression data derived from a meta-analysis of 31624 blood samples collated by the eQTLGen consortium[26]. Thirteen genes were annotated to within 1 Mb of the chromosome

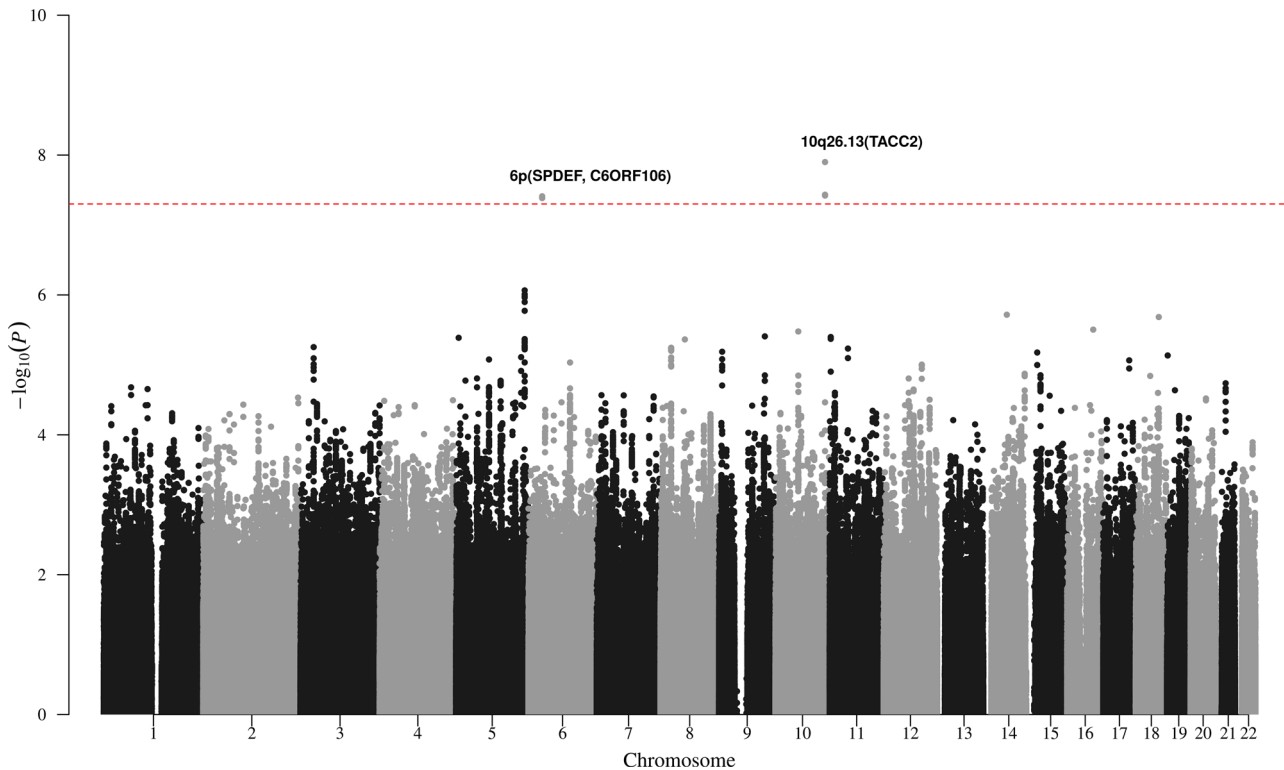

**Fig. 1 Manhattan plot from chronic lymphocytic leukemia meta-analysis of 6 genome-wide association studies for variants associating with progressive disease.** Manhattan plot of fixed-effects meta-analysis results for time to first treatment (TTFT). Study-specific single nucleotide polymorphism (SNP) effect sizes were combined using the inverse-variance-weighted approach, and nominal meta $P$ values were calculated based on $Z$-score statistics. The $x$-axis represents SNP coordinates on autosomal chromosomes and the $y$-axis shows the fixed-effects meta $P$ values in the $-\log_{10}$ scale. All statistical tests were two-sided and red dashed line indicates significance at genome-wide level ($P \leq 5 \times 10^{-8}$). Risk loci for progressive CLL are annotated with chromosome position and local genes.

10 association signal and rs736456 is eQTL for the *PLEKHA1* gene (pleckstrin homology domain-containing family A; TAPP-1) (Benjamini-Hochberg corrected $P$-value [$P_{BH}$] $= 1.29 \times 10^{-15}$) (Supplementary Table 2). Of the 27 genes annotated to within 1 Mb of the chromosome 6 signal, rs3778076 is eQTL for 5 genes including most significantly *UHRF1BP1* (Ubiquitin-Like Containing PHD And RING Finger Domains 1-Binding Protein 1) ($P_{BH} = 6.73 \times 10^{-139}$) and *ILRUN* (Inflammation And Lipid Regulator With UBA-Like And NBR1-Like Domains; *C6ORF106*) ($P_{BH} = 3.54 \times 10^{-64}$)(Supplementary Table 3).

**Shared risk alleles for CLL etiology and progressive disease.** rs3778076 maps close (103 kb) to an established CLL etiological risk variant on chromosome 6q21 (rs3800461)[20] and the two variants are in linkage disequilibrium (*r2* 0.46, D' 0.91) suggesting they might drive initial CLL development and subsequent progression to symptomatic disease via a common mechanism. Consistent with this hypothesis, rs3800461 is also an eQTL for *UHRF1BP1* and *ILRUN* (*C6ORF106*) (Supplementary Table 4), although we cannot exclude these variants operating through independent mechanisms. Of other CLL risk variants identified via large scale GWAS[18–25], data on 5 variants did not pass quality control in our study (rs9815073, rs6858698, rs10069690, rs7944004, rs71597109) and none of the others were significantly ($P < 10^{-3}$) associated with progressive disease (Supplementary Figs. 18–59).

**Discussion**
CLL susceptibility loci identified to date implicate genes central to B-cell development, BCR signaling and apoptotic responses[20]. These risk alleles may therefore function by promoting the

generation of abnormal pro-B populations or to maintain the outgrowth of CLL-precursors in response to continuous BCR signaling. However, the current model would suggest that once the pre-leukemic CLL clone has established, disease progression becomes largely dependent on the classic multi-hit model of oncogenesis where acquired somatic drivers promote clonal expansion[11], thus temporally limiting the impact of constitutional genetic variants to disease progression at an early stage.

The evidence presented herein suggests that, in addition to being etiological, constitutional risk alleles contribute to disease progression in CLL post-diagnosis. Our data suggest that rs736456 and rs3778076 are powerful prognostic markers in early stage CLL, but have less impact as disease progresses, predicted to be due to the acquisition of strongly prognostic somatic alterations which then dominate. Consistent with this model, the prognostic power of rs3778076 and rs735456 is weaker in patients with Binet stage B or C disease and in patients with high risk somatic markers, such as unmutated *IGHV* or CD38 positivity.

The progressive disease allele at chromosome 10 (rs736456) is associated with increased expression of *PLEKHA1* (pleckstrin homology domain containing A1; TAPP1) whilst the allele at chromosome 6 (rs3778076) is associated with increased expression of *ILRUN* (Inflammation And Lipid Regulator With UBA-Like And NBR1-Like Domains; *C6ORF106*) and *UHRF1BP1* (UHRF binding protein 1). PLEKHA1, a plekstrin homology (PH) adapter protein is a target for phosphatidylinositol-4,5-bisphosphate 3-kinase (PI3K) signaling and a partner for phosphatidylinositol 4,5-bisphosphate (PIP2) which regulates the assembly of signaling complexes at the cell membrane[27]. PI3K-mediated signaling is a key element of BCR-mediated signal transduction, a receptor with a central, etiological role in CLL

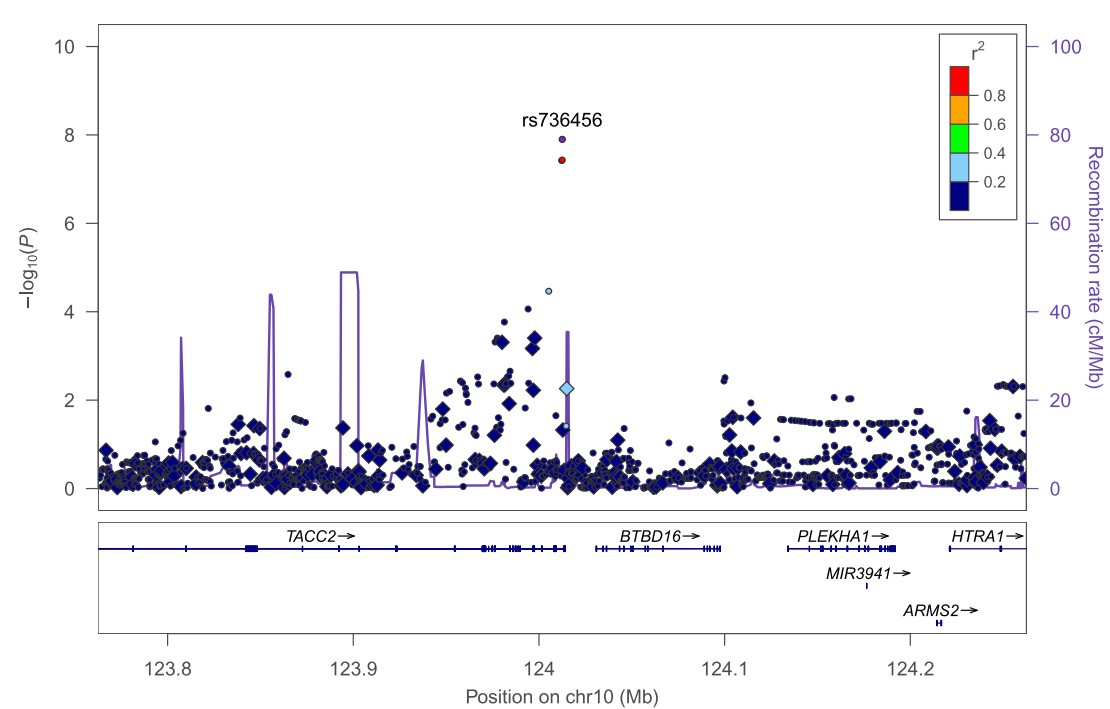

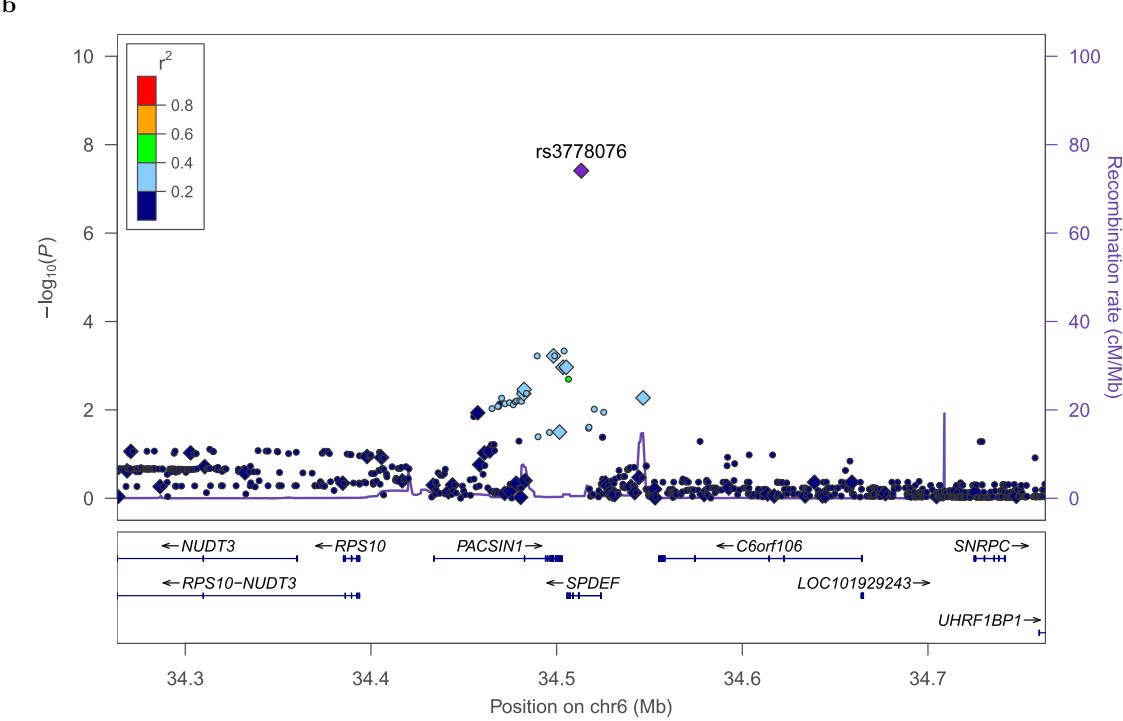

**Fig. 2 Regional association and linkage disequilibrium plots for loci associated with progressive CLL.** Regional plots of time to first time treatment (TTFT) survival associations for rs736456 (**a**) and rs3778076 (**b**) reaching genome-wide significance ($P \le 5 \times 10^{-8}$). Study-specific single nucleotide polymorphism (SNP) effects were combined in a fixed effect model using an inverse-variance-weighted method. Nominal meta-analysis $P$ values were calculated based on $Z$-score statistics and all statistical tests were two-sided. Plots show SNP coordinates based on genomic build b37/hg19 on the $x$-axis, and $-\log_{10}$ ($P$ values) on the $y$-axis. Genotyped and imputed SNPs are represented by diamonds and circles, respectively, and are colored according to their linkage disequilibrium (pairwise $r^2$) with the lead SNP based on the 1000 Genomes European panel. Reference genes in the region are shown in the lower panel, with arrows indicating transcript direction, dense blocks representing exons and horizontal lines representing introns. Two additional SNPs at 10q26.13 (rs4752676 and rs736457) (**a**) and one additional SNP at 6p (rs11757517) (**b**) also reach genome-wide significance in meta-analysis (Supplementary Figs. 60–62). The symbol for rs11757517 is obscured on the regional plot (**b**) by the symbol for rs3778076.

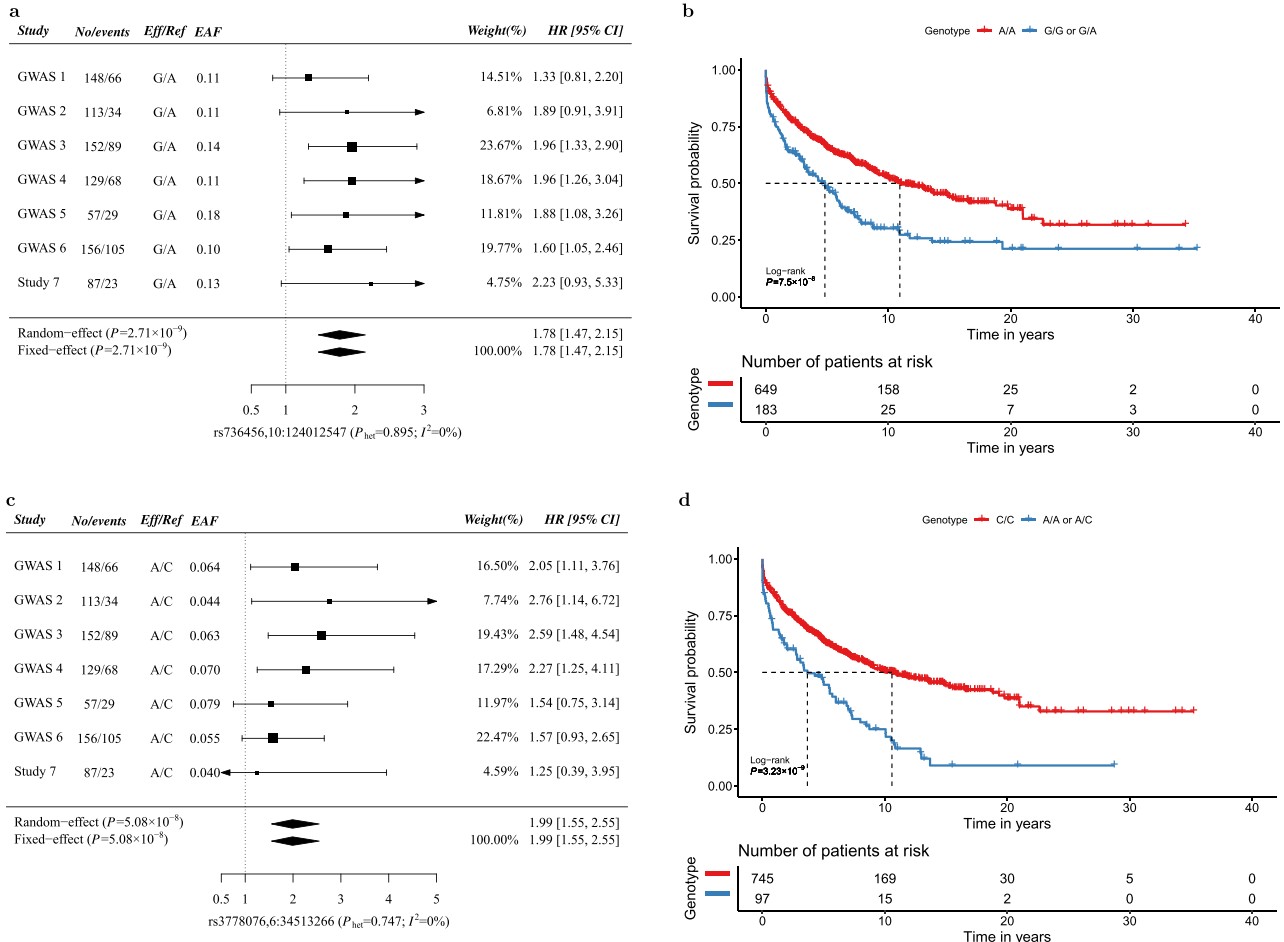

**Fig. 3 Forest plots and survival curves for loci associated with progressive CLL.** Panels **a** and **c** show forest plots for rs736456 (**a**) and rs3778076 (**c**) for the association with time to first treatment (TTFT) by GWAS. No/events: Number of CLL patients/Number of patients receiving treatment; Eff/Ref: effect/reference allele; EAF: effect allele frequency; HR: hazard ratio; CI: confidence interval; For each study, allelic dosage was estimated for SNP and included in a Cox proportional hazard model to estimate HR and 95% CI. Wald-type CIs based on the standard normal distribution were reported here. Squares denote the per-allele HR, with size proportional to the weight of the study. Pooled HRs derived from both the fixed and random-effects models are indicated by diamonds with their corresponding meta $P$ values from Wald tests shown in the left parentheses. $X$-axis label formats include reference sequence (rs) identifier and chromosome:position (b37). $P$ values for Cochran's Q test ($P_{het}$) and $I^2$ for heterogeneity are shown in parentheses. Panels **b** and **d** show Kaplan–Meier survival plots for TTFT in CLL patients stratified by rs736456 (**b**) and rs3778076 (**d**). TTFT is defined as the time from diagnosis of CLL to treatment or last follow-up without treatment. The number of patients in each group at each timepoint are indicated in the table. Nominal $P$ values from log-rank tests are shown. All statistical tests were two-sided.

pathogenesis[28]. Herold and colleagues identified and validated an 8-gene expression signature including *PLEKHA1* that predicted time to first treatment in CLL[29]. Furthermore, the PLEKHA1 homologous protein TAPP2 is differentially expressed in CLL, with increased expression in ZAP70 positive/*IGHV* unmutated CLL, a subtype associated with enhanced signaling via the BCR with associated pro-survival effects on cultured CLL-cells[30,31].

ILRUN (C6ORF106) regulates interferon regulatory factor 3 (IRF3) signaling which modulates cellular responses to viral antigens. Given that CLL may arise in response to chronic stimulation of the BCR by viral antigens, constitutional regulation of an IRF3-dependent response may therefore affect the expansion of CLL-clones[32]. IRF3 also regulates expression of miR-155[33,34] which is thought to be etiological in a subset of patients[33,35] and also associates with unmutated *IGHV* and aggressive disease in CLL[36–40].

*UHRF1BP1* has been identified as a susceptibility gene for systemic lupus erythematosus (SLE)[41] with differential methylation affecting regulation of this gene[42], however UHRF1BP1 has not previously been associated with B-cell tumors. An etiological

CLL GWAS identified susceptibility variants mapping to within the B-cell scaffold protein with ankyrin repeats 1 (BANK1) gene[20]. BANK1 is an adapter for BCR signals, modulates CD40 receptor-mediated signaling responses and has also been implicated in SLE and other inflammatory arthropathies[43–45]. Therefore, the association of both UHRF1BP1 and BANK1 with CLL and SLE could imply common etiological factors between the two diseases. Given that both CLL and SLE are considered to arise, in part, due to a loss of immunological tolerance to self-antigens or auto-antigens this may provide further indirect evidence for auto-antigenic drive in the etiology of CLL.

Collectively the risk alleles identified herein map to genes known to be regulatory for B-cell development and the control of autoimmunity supporting the concept that either aberrant BCR signals and/or a breakdown in normal immunological tolerance contributes to CLL clonal expansion and the subsequent acquisition of somatic driver mutations.

Many patients with Binet stage A CLL never progress to the point of therapeutic intervention and the accurate identification of patients in this group has been the subject of intensive

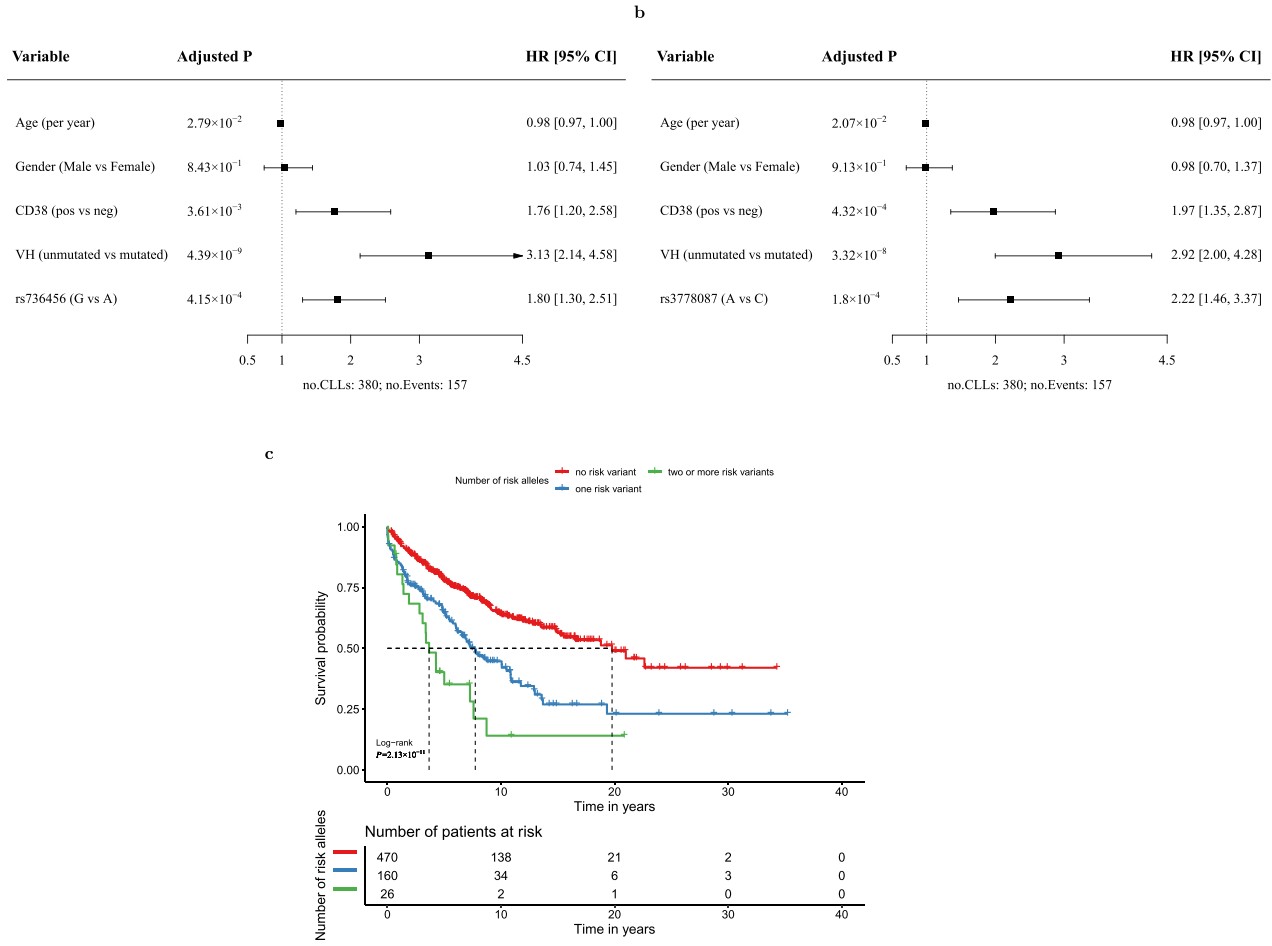

**Fig. 4 Multivariate prognostication models for time to first treatment (TTFT) and survival curves in Binet stage A CLL.** Study-stratified multivariable Cox proportional hazard model for TTFT in Binet A CLL patients incorporating rs736456 (**a**) and rs3778076 (**b**). Age at diagnosis (continuous), gender (binary), CD38 (binary), VH (binary) and SNP allelic dosage for the effect allele (continuous) were included together in a study-stratified Cox proportional hazard model to estimate Hazard ratios (HR) and 95% confidence intervals (CI). CI and nominal *P* values from Wald tests based on the standard normal distribution were reported here. Cases with missing data for any of the variables were excluded from multivariate models. **c** Kaplan–Meier survival plot for TTFT by number of cumulative risk alleles (0, 1, >1) for rs3778076 and rs736456 in Binet A CLL patients. TTFT is defined as the time from diagnosis of CLL to treatment or last follow-up without treatment. The number of patients in each group at each timepoint are indicated in the table. Nominal *P* value from the log-rank test is shown. All statistical tests were two-sided.

investigation. Multiple prognostic features and tests have been described to identify those with a low risk of progressive disease. Conversely, the identification of high-risk prognostic features in stage A CLL patients has been used to define a group likely to need close surveillance as opposed to community monitoring programmes. The failure, to date, to demonstrate a survival benefit for early treatment of high-risk asymptomatic patients is due to an inability to successfully identify an appropriate high-risk group and also the limits of pre-emptive chemotherapy-based approaches[46]. However, with the advent of highly-effective targeted therapies such as BCRi pre-emptive therapy in high-risk CLL patients is again under investigation[47].

Genetic tests for constitutional genetic markers, such as rs736456 and rs3778076, have the advantage of being easy to perform, highly reproducible and inexpensive making them ideal for incorporation into multivariate prognostication models for disease progression in early stage CLL.

## Methods

**Patients and ethics**. CLL patients were diagnosed at clinical centers in the United Kingdom, including Bournemouth (Royal Bournemouth Hospital, GWAS 1), Cardiff (University Hospital of Wales, GWAS 2), Hull (Hull University Teaching Hospital NHS Trust, GWAS 3 and GWAS 4, study 7), Newcastle (Royal Victoria

Infirmary, GWAS 5) and Gateshead/Sunderland (Queen Elisabeth Hospital/City Hospitals Sunderland NHS Trust, GWAS 6). Collection of patient samples and associated clinico-pathological information was undertaken with written informed consent. All studies were conducted in accordance with the Declaration of Helsinki and received local institutional review board and/or research ethics approval (Hull 08/H1304/35; Cardiff 02/4806). Newcastle, Gateshead, and Sunderland CLL patient samples were collected and stored following approval from the Newcastle Academic Health Partners Biobank (http://www.ncl.ac.uk/biobanks/collections/nbrtb/).

**CLL diagnosis, CD38/IGHV analysis**. All patients had typical CLL with an absolute B cell count >5 × 109 L$^{-1}$ expressing CD5, CD23, CD19 and were diagnosed in accordance with World Health Organisation guidelines[48] contemporaneous to the time of diagnosis. Cases were recruited to each study in the following time periods: GWAS1 (1970–2005), GWAS2 (1979–2009), GWAS3 (1970–2009), GWAS4 (1997–2013), GWAS5 (1972–2002), GWAS6 (1969–2015), study 7 (1997–2017). *IGHV* gene sequences were analyzed using IgBlast (v1.3.0) (https://www.ncbi.nlm.nih.gov/igblast/) and those with at least 98% homology to germline were regarded as unmutated[10]. CD38 expression was determined at each individual clinical center via flow cytometric analysis of diagnostic samples and a cut-off of 30% was used for positivity[49]. Serum β2 microblobulin was performed by referring centers using locally established and validated immunoassays. Assessment of deletion 17p and *TP53* mutation analysis was in line with ERIC recommendations on *TP53* mutation analysis in chronic lymphocytic leukemia[50].

Treatment was initiated in response to standard indications[51]. Given the historic nature of the treated cohorts in this study the treatment regimens employed were predominantly chemotherapeutic (chlorambucil monotherapy, fludarabine monotherapy, fludarabine, and cyclophosphamide) or chemoimmunotherapeutic

(chemotherapy in combination with a CD20 monoclonal antibody). A minority of patients were treated with other regimens including radiotherapy, alemtuzumab, ibrutinib or ibrutinib in combination with venetoclax.

Unmutated *IGHV*, CD38-positivity, Binet stage B or C, serum β2 microbulin >3.5 mg L$^{-1}$ and deletion/mutation of *TP53* at presentation were all significantly associated with shorter time to first treatment (TTFT) in univariate analysis (Supplementary Figs. 1–5; $P < 0.05$), confirming that established prognostic markers predict disease progression in patients recruited to this study.

**Genotyping and quality control procedures**. DNA was extracted from peripheral blood and genotyped on Illumina OmniExpress arrays (8v1, 8v1-2, or 8v1.3). Genotypes were determined using Illumina GenomeStudio software (v2.0) (Illumina) with 922269 SNPs common to all three arrays (Supplementary Fig. 6). Quality control and data analysis were performed using R v3.5.1 and PLINK v1.9b4.4 (Supplementary Fig. 6). Given that CLL cases were genotyped using DNA extracted from tissue samples that include leukemic B cells we applied rigorous quality control to remove variants potentially affected by somatic copy number alterations. For each GWAS we excluded markers with departure from Hardy–Weinberg equilibrium (HWE; $P \le 10^{-3}$), a call rate <95% or with significant differences in minor allele frequency ($P \le 10^{-3}$) between genotype batches. Data from all six GWAS were combined for sample quality control processing. Samples were excluded if the call rate was < 95%, heterozygosity exceeded 3 standard deviations from the overall mean heterozygosity or were identified as non-European based on principal components analysis using 1000 genome data as a reference (Supplementary Fig. 7). Samples were also removed so that there were no two individuals with estimated relatedness pihat ≥0.1875, with retention of the sample with the higher call rate. As an additional control check to ensure that the constitutional variants reported here are not located within recurrent regions of somatic copy number change we used Nexus Copy Number software 10 (build version 9665) (BioDiscovery, California) to interrogate B allele frequency and Log R ratio at variant positions.

Following the exclusion of poor quality markers and samples, haplotypes were estimated from genotypes using ShapeIT (v2.r790)[52] and genome-wide imputation was performed using the Michigan Imputation Server (https://imputationserver.sph.umich.edu/index.html) and the Haplotype Reference Consortium reference haplotype panel (http://www.haplotype-reference-consortium.org/). All variants with an imputation info score <0.9 were excluded from subsequent analysis.

**Genome-wide analysis for markers that predict time to first treatment**. The primary outcome assessed in this study was time to first treatment (TTFT), defined as the interval between CLL diagnosis and date of first treatment or last follow-up. For each study, allelic dosage was estimated for the minor allele at each variant position and included in a cox proportional hazard model to estimate hazard ratio (HR) and 95% confidence interval (CI). Variants were included in the meta-analysis if they had results from all six studies. Study-specific single nucleotide polymorphism (SNP) effects were combined using an inverse-variance-weighted method (fixed effects model) and the DerSimonian–Laird approach (random effects model) using R metafor package (v2.4-0)[53]. SNPs with fixed-effect *P* values of ≤5 × 10$^{-8}$ were deemed significant at genome-wide level. Cochran's Q test and $I^2$ were used to assess the heterogeneity across studies. Regional plots were generated using LocusZoom (V1.4) downloaded from https://github.com/statgen/locuszoom-standalone.

To test for secondary signals in regions achieving genome-wide significance we carried out conditional analysis on the index SNP by study and then combined individual conditional effect sizes from 6 studies under a fixed effects model. SNPs with a conditional meta *P* value <10$^{-4}$ were considered independent of the index SNP.

Heterogeneity of Kaplan-Meier survival curves was assessed by the log-rank test, including tests for established prognostic markers, the most likely index SNP genotypes (for genotypes with posterior probabilities >0.9), combined groups of established prognostic markers/SNPs and combined SNP genotypes. For joint analyses, pairwise log-rank tests, with false discovery rate (FDR) corrections for multiple testing, were used to test for significant differences between survival curves.

Study-stratified multivariate Cox proportional hazard models incorporating established prognostic markers including *IGHV* and CD38 status were used to determine the impact of variants on disease progression. Cases with missing data for any of the variables were excluded from multivariate models.

A secondary outcome was post-treatment survival, defined as the interval between first treatment for CLL-related symptoms and death or last follow-up. Data on post-treatment survival were available on 390 CLL cases from six studies (GWAS1-6), with 231 deaths and 159 censored at last follow-up.

**Technical validation of variants associating with progressive CLL**. rs3778076 was directly genotyped in all six GWAS and rs736456 was imputed to high quality in all six GWAS (info score 0.974–0.985). Fidelity of array genotyping and imputed dosages was confirmed using Sanger sequencing in a subset of samples for each sentinel variant with concordance of 100% for all variants (Supplementary Fig. 8). rs3778076 and rs736456 were directly genotyped in study 7 using Sanger sequencing.

**Expression quantitative trait loci analysis**. To identify *cis* expression quantitative trait loci (eQTLs) we made use of summary data from the eQTLGen Consortium for whole blood, which incorporates 37 independent datasets derived from a total of 31684 individuals (http://www.eqtlgen.org/cis-eqtls.html)[26]. Benjamini-Hochberg (BH)-adjusted *P* values were estimated for each gene annotated to within 1 Mb of rs736456 on 10q26.13 and rs3778076 on 6p.

**Reporting summary**. Further information on research design is available in the Nature Research Reporting Summary linked to this article.

## Data availability
Genome-wide association summary statistics (Lin_TTFT_CLLmetaAssoc.txt) are available for download from https://doi.org/10.25405/data.ncl.12136365. eQTL data is available from the eQTLGen consortium via http://www.eqtlgen.org/cis-eqtls.html (2019-12-11-cis-eQTLsFDR-ProbeLevel-CohortInfoRemoved-BonferroniAdded.txt.gz). The remaining data are contained within the supplementary information files or available from the authors upon reasonable request. Source data are provided with this paper.

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

## Acknowledgements

This work was funded by Blood Cancer UK (formerly Bloodwise) (#06002 and #13044 to J.M.A.). J.M.A. is recipient of an Oncology/Haematology Fellowship from Gilead Pharmaceuticals and C.E. received funding from JGW Patterson Foundation. This work was supported by Cancer Research UK (C2115/A21421). E.W. and G.S. received funding from Blood Cancer UK (#14024). A.S. was partly funded by the National Institute for Health Research Oxford Biomedical Research Centre; the views and opinions expressed are those of the authors and do not necessarily reflect those of the National Institute for Health Research, the UK National Health Service, the UK Department of Health or the Universities of Oxford. We are grateful for the support of the Clinical Trials Research Unit at The University of Leeds and the Hull York Medical School, University of Hull for funding the Daisy Tumour Bank.

## Author contributions

W.-Y.L. collated data, conducted data analysis and drafted the manuscript. S.E.F., N.S., C.E., T.R., E.W., C.S., G. Strathdee, T.M.-F., R.P., H.Mearns, T.B., R.S.H., H.Marr, J.W., G.Summerfield, S.M., A.P., C.P., C.F., F.F., M.J.S.D., S.J., A.Sellors, A.Schuh, P.R., D.O., J.B., S.R., A.B., L.C., P.E., P.H., and G.P. generated/collated data and/or advised on data analysis. J.M.A. and D.J.A. designed the project, generated data, analyzed data, directed the research and drafted the manuscript. J.M.A. also obtained funding. All authors contributed to the final version of the manuscript.

## Competing interests

The authors declare no competing interests.
