## [Peer Review File · Nature Communications]

Editorial Note: Reviewer #2 was unavailable to comment on the revised manuscript and Reviewers #3 and #4 were recruited to comment on the author's response to Reviewer #2.

Reviewers' comments:

Reviewer #1 (Remarks to the Author):

The authors proposed a genome wide association study (GWAS) for time to first treatment (TTFT) in a pooled cohort of chronic lymphocytic leukemia patients (CLL) with the hypothesis that inherited variants affected CLL prognosis. The authors reported two novel loci associated with TTFT, especially among those CLL cases with good prognostic factors.

In general, the methods are lacking many details. Some may affect the conclusions. Regardless the authors failed to show that these SNPs provide prognostic value above that when one jointly accounts for the known CLL factors (e.g., using the CLL IPI). Thus limiting the significance of the findings.

What were the diagnosis year of the CLL patients? What was the distribution of absolute B-cell count? Given older treatment regimens were included herein, there may be a number of CLL who did not meet the diagnostic WHO criteria and as such be classified as MBLs. Further, supplementary figure 10 showing histograms of sample characteristics is not helpful. A table should be provided along with testing for any difference across studies as well as indicator of missing data.

Methods are lacking many details:

No details from where DNA was obtained. Presumably blood. When was the blood sample collected or how far out from CLL diagnosis was the blood sample collected? If at time of therapy, then DNA from whole blood may be contaminated by tumor. Analyses should be from time of sample collection rather than dx date.

Need to correct text (page 7, line 123) stating that different arrays were used for genotyping the CLLs as shown in supplemental fig 7.

Lacking imputation details in Methods. Supplemental fig 7 seems to imply that each array was reduced to common SNPs across the arrays (n=922269) and then imputation was done. Unclear whether sample were excluded prior to imputation or after?

The analyses should adjust for study

No details in Methods about the analyses for post-treatment survival (line 184, page 10). Given the reduce number of samples in the survival analyses, which study had survival data? Was treatment taken into account in the analyses? What about age?

No details in Methods about eQTL analyses.

The analyses looking at the top SNPs and Binet stage is unclear. Was stratified analyses performed? Supplementary Fig 14 is not standard to show two p-values within KM plot. Instead, you have two KM plots for each stratum. Was a test for interaction performed? Likewise for Supp figures 15 and 16

Figure 1 should be moved to supplemental.

The results are interesting and the authors show consistency across the studies. The authors conclude that the novel SNPs improve prognostication especially among those with good prognostic markers. But the authors did not jointly look among all the prognostic markers. Only one at a time. Given that these prognostic markers are typically performed in clinical practice, it is important to show that these SNPs provide information above and beyond what is already currently obtained in practice. E.g, using CLL-IPI low risk or stratified using only cases that are mutated IGVH, Binet stage A, and CD38 negative.

Reviewer #2 (Remarks to the Author):

This is an interesting paper reporting a GWAS meta-analysis focused on time to first therapy in CLL patients. Perhaps the most interesting feature is that the SNPs seem most predictive among lower risk patients, i.e. those with Binet A disease or mutated IGHV, in whom we need better markers. I have several concerns: (1) ancestry variation among the patients and cohorts, particularly given 2 of the cohorts seem to separate from the others in the PCA; (2) inadequate consideration of prognostic variables important in CLL, in particular FISH cytogenetics is not presented anywhere or included in univariate much less multivariable models; these data are essential and the new SNPs need to be found to be independent of higher risk FISH; (3) a lot of missing IGHV data which is perhaps the strongest predictor of TTFT; (4) focus on CD38 which is never significant in multivariable models for CLL outcome when these other factors are included; (5) it is not clear why the SNPs would predict TTFT only in Binet A, but not B and C patients. C patients need therapy right away, but the SNPs should retain prognostic significance in Binet B; (6) Given there is no control group, I think an independent validation cohort demonstrating that these SNPs still have independent prognostic significance for TTFT is needed. There is a long history of findings that do not hold up in validation cohorts.

Other points:

- the discussion of prognostic markers in the Introduction is very naïve from a clinical CLL point of view, should be better prioritized. In particular, CD38 can vary from day to day within a patient, and is not a stable prognostic marker, so I would not focus so much on it.
- supplementary figure 8 – the colors for the ancestries are not visible and neither are all the GWAS visible. Looks like GWAS 1 and 2 are separated from the others.
- excluding non-Europeans based on PCA analysis using 1000 genome data is not really adequate control for ancestry variation, particularly as the PCA suggests some ancestry heterogeneity.
- similarly the comment that CD38 positive disease along with unmutated IGHV and Binet B or C disease predict shorter TTFT and OS in univariate analysis is inadequate to really address this cohort. We need a full report of all factors associated with TTFT and OS in univariate analysis, as well as multivariate analysis. It furthermore appears that there are a LOT of missing data on IGHV, which has to be very well controlled for in this analysis. How was missing IGHV data handled in the multivariable analysis?
- 5 variants look separated on the QQ plot but only 2 come up in the analysis? What are the others and how do they look?
- the Discussion states the CLL is a precursor B cell neoplasm; this is incorrect, it is a mature B cell neoplasm. In addition the first half of the Discussion is mostly pure speculation about CLL causation that has nothing to do with this paper and should be modified.

Lin et al, Genome-wide association study identifies risk loci for progressive chronic lymphocytic leukemia

We thank the reviewers for their excellent comments. We have addressed each of the reviewer's comments as follows:

Reviewer #1 (Remarks to the Author):

Reviewer 1 comment 1: The authors proposed a genome wide association study (GWAS) for time to first treatment (TTFT) in a pooled cohort of chronic lymphocytic leukemia patients (CLL) with the hypothesis that inherited variants affected CLL prognosis. The authors reported two novel loci associated with TTFT, especially among those CLL cases with good prognostic factors.

In general, the methods are lacking many details. Some may affect the conclusions. Regardless the authors failed to show that these SNPs provide prognostic value above that when one jointly accounts for the known CLL factors (e.g., using the CLL IPI). Thus limiting the significance of the findings.

Author response: The reviewer has elaborated on these points in comment 12 - please see our response to comment 12.

Reviewer 1 comment 2: What were the diagnosis year of the CLL patients? What was the distribution of absolute B-cell count? Given older treatment regimens were included herein, there may be a number of CLL who did not meet the diagnostic WHO criteria and as such be classified as MBLs.

Author response: All cases were diagnosed with CLL based on World Health Organisation criteria contemporaneous to the time of diagnosis. Given that diagnostic techniques for CLL have become progressively more sensitive over time it is unlikely that our study includes any MBL cases, although we cannot exclude this possibility. We have included new text to the Methods section to clarify these points along with dates of diagnosis for each study, as follows:

"All patients had typical CLL with an absolute B cell count $>5 \times 10^9 \text{ L}^{-1}$ expressing CD5, CD23, CD19 and were diagnosed in accordance with World Health Organisation guidelines contemporaneous to the time of diagnosis. Cases were recruited to each study in the following time periods: GWAS1 (1970-2005), GWAS2 (1979-2009), GWAS3 (1970-2009), GWAS4 (1997-2013), GWAS5 (1972-2002), GWAS6 (1969-2015), study 7 (1997-2017). *IGHV* gene sequences were analysed using IgBlast (<https://www.ncbi.nlm.nih.gov/igblast/>) and those with at least 98% homology to germline were regarded as unmutated. CD38 expression was determined at each individual clinical centre via flow cytometric analysis of diagnostic samples and a cut-off of 30% was used for positivity. Serum $\beta 2$ microglobulin was performed by referring centres using locally established and validated immunoassays. Assessment of deletion 17p and *TP53* mutation analysis was in line with ERIC recommendations on *TP53* mutation analysis in chronic lymphocytic leukemia²⁸."

Reviewer 1 Comment 3: Further, supplementary figure 10 showing histograms of sample characteristics is not helpful. A table should be provided along with testing for any difference across studies as well as indicator of missing data.

Author response: We have converted supplementary Figure 10 to a Table (now included as Supplementary Table 1) and included statistical analysis to test for significant differences between the cohorts.

Reviewer 1 comment 4: Methods are lacking many details:

No details from where DNA was obtained. Presumably blood. When was the blood sample collected or how far out from CLL diagnosis was the blood sample collected? If at time of therapy, then DNA from whole blood may be contaminated by tumor. Analyses should be from time of sample collection rather than dx date.

Author response: DNA was extracted from peripheral blood in all cases. The exact date of sampling is not known for many of the cases recruited to our study. For the majority of cases where the date of sampling is known, peripheral blood was taken at the time of CLL diagnosis or within a few years of diagnosis. For a small number of cases, sample collection occurred more than 5 years after diagnosis. Regardless, the reviewer is correct in that there will likely be some somatic cell content in DNA samples. As such, we apply extremely rigorous quality control metrics in order to remove variants and samples significantly affected by somatic copy number alterations and to ensure high quality data. Specifically, we routinely exclude variants with departure from Hardy-Weinberg equilibrium ($P < 10^{-3}$) and with a low call rate ($< 95\%$), indicative of variants within somatically deleted/amplified regions. We also exclude variants that show significant differences ($P < 10^{-3}$) between genotype batches. Moreover, individual samples with a call rate of $< 95\%$ or with high or low heterozygosity rates (± 3 standard deviation from the mean) are also excluded – low heterozygosity is a common feature of regions affected by somatic deletion or copy neutral loss of heterozygosity.

As an additional control check we also use Nexus Copy Number software (BioDiscovery, California) to interrogate B allele frequency and Log R ratio at variant positions. We use this approach as a final check to ensure that the constitutional variants we are reporting are not located within recurrent regions of somatic copy number change in CLL.

We have added some text to the methods section of our revised manuscript to cover these points, as follows:

“DNA was extracted from peripheral blood and genotyped on Illumina OmniExpress arrays (8v1, 8v1-2 or 8v1.3). Genotypes were determined using Illumina GenomeStudio software with 922269 SNPs common to all three arrays (Supplementary Figure 6). Quality control and data analysis were performed using R v3.5.1 and PLINK v1.9b4.4 (Supplementary Figure 6). Given that CLL cases were genotyped using DNA extracted from tissue samples that include leukemic B cells we applied rigorous quality control to remove variants potentially affected by somatic copy number alterations. For each GWAS we excluded markers with departure from Hardy-Weinberg equilibrium (HWE; $P \leq 10^{-3}$), a call rate $< 95\%$ or with significant differences in minor allele frequency ($P \leq 10^{-3}$) between genotype batches. Data from all six GWAS were combined for sample quality control processing. Samples were excluded if the call rate was $< 95\%$, heterozygosity exceeded 3 standard deviations from the overall mean heterozygosity or were identified as non-European based on principal components analysis using 1000 genome data as a reference (Supplementary Figure 7). Samples were also removed such that there were no two individuals with estimated relatedness $\text{pihat} \geq 0.1875$, with retention of the sample with the higher call rate. As an additional control check to ensure that the constitutional variants reported here are not located within recurrent regions of somatic copy number change we used Nexus Copy Number software (BioDiscovery, California) to interrogate B allele frequency and Log R ratio at variant positions.”

Reviewer 1 comment 5: Need to correct text (page 7, line 123) stating that different arrays were used for genotyping the CLLs as shown in supplemental fig 7.

Author response: We thank the reviewer for highlighting this omission and have corrected the manuscript text accordingly, as follows:

“DNA was extracted from peripheral blood and genotyped on Illumina OmniExpress arrays (8v1, 8v1-2 or 8v1.3). Genotypes were determined using Illumina GenomeStudio software with 922269 SNPs common to all three arrays (Supplementary Figure 6).”

Reviewer 1 comment 6: Lacking imputation details in Methods. Supplemental fig 7 seems to imply that each array was reduced to common SNPs across the arrays ($n=922269$) and then imputation was done. Unclear whether sample were excluded prior to imputation or after?

Author response: Elimination of poor quality samples was done prior to imputation, and the number of SNPs ($N=504370$) retained for imputation is now noted in supplementary figure 6. We have also included some text in the Materials and Methods to clarify this point, as follows:

Materials and Methods:

“Following the exclusion of poor quality markers and samples, haplotypes were estimated from genotypes using ShapeIT (v2.r790) and genome-wide imputation was performed using the Michigan Imputation Server (<https://imputationserver.sph.umich.edu/index.html>) and the Haplotype Reference Consortium reference haplotype panel (<http://www.haplotype-reference-consortium.org/>). All variants with an imputation info score < 0.9 were excluded from subsequent analysis.”

Reviewer 1 comment 7: The analyses should adjust for study

Author response: Analysis was performed using a Cox proportional hazards model stratified by study. We have now included additional text in the Results and the Figure 4 legend to make clear that the results are adjusted for study, as follows:

Results:

“Both genetic markers retained significance in multivariate models for disease progression restricted to Binet stage A patients and when adjusted for study in the model (Figure 4).”

Legend to Figure 4:

“Hazard ratios (HR) and 95% confidence intervals (95% CI) for each variable were estimated after adjustment for study and other variables in the model.”

Reviewer 1 comment 8: No details in Methods about the analyses for post-treatment survival (line 184, page 10). Given the reduce number of samples in the survival analyses, which study had survival data? Was treatment taken into account in the analyses? What about age?

Author response: Post-treatment survival data was available from six studies (GWAS1-6). All patients included in these analyses had received treatment for CLL-related symptoms with the results presented by study and adjusted for age. We have included additional text on the post-treatment survival analysis, as follows:

“A secondary outcome was post-treatment survival, defined as the interval between first treatment for CLL-related symptoms and death or last follow-up. Data on post-treatment survival were available on 390 CLL cases from six studies (GWAS1-6), with 231 deaths and 159 censored at last follow-up.”

We have also modified the legend to Supplementary Figure 11, as follows:

Supplementary Figure 11: Forest plots showing associations between post-treatment survival and risk variants for progressive CLL. Forest plots for rs736456 (A) and rs3778076 (B) and their age-adjusted association with post-treatment survival stratified by GWAS. Post-treatment survival is defined as the time from first treatment for CLL-related symptoms to death or last follow-up. No/events: Number of CLL patients/Number of patients receiving treatment; Eff/Ref: effect/reference allele; EAF: effect allele frequency; HR: hazard ratio; CI: confidence interval; Squares denote the per-allele HR, with size proportional to the weight of the study. Pooled HRs derived from both the fixed and random-effects models are indicated by diamonds with their corresponding meta *P* values shown in the left parentheses. X-axis label formats include reference sequence (rs) identifier and chromosome:position (b37). *P* values for Cochran’s *Q* test (P_{het}) and I^2 for heterogeneity are shown in parentheses.”

Reviewer 1 comment 9: No details in Methods about eQTL analyses.

Author response: We have included additional text in the Methods describing the eQTL analysis, as follows:

“Expression quantitative trait loci analysis.

To identify *cis* expression quantitative trait loci (eQTLs) we made use of summary data from the eQTLGen Consortium for whole blood, which incorporates 37 independent datasets derived from a total of 31684 individuals (<http://www.eqtlgen.org/cis-eqtls.html>)³². Benjamini-Hochberg (BH)-adjusted *P* values were estimated for each gene annotated to within 1Mb of rs736456 on 10q26.13 and rs3778076 on 6p.”

Reviewer 1 comment 10: The analyses looking at the top SNPs and Binet stage is unclear. Was stratified analyses performed? Supplementary Fig 14 is not standard to show two p-values within KM plot. Instead, you have two KM plots for each stratum. Was a test for interaction performed? Likewise for Supp figures 15 and 16

Author response: *P* values were obtained from pairwise log-rank tests between survival curves with false discovery rate corrections for multiple testing. We have retained the analyses for Binet stage by genotype on a single plot for ease of comparison (Supplementary Figure 12). We have applied the same approach to the analyses for *IGHV*, CD38, β 2 microglobulin and TP53 status (Supplementary Figures 13-16). We have included additional text describing the analysis used to supplementary figures 12-16, as follows:

“Supplementary Figure 12: Kaplan-Meier plot showing time to first treatment (TTFT) for CLL patients stratified by Binet stage of disease and SNP genotype for rs736456 (A) and rs3778076 (B). *P* values are obtained from pairwise log-rank tests for survival curves, with false discovery rate (FDR) corrections for multiple testing. For simplicity, *p* values are only shown for the comparisons of SNP genotypes within the same prognostic factor stratum. TTFT is defined as the time from diagnosis of CLL to treatment or last follow-up without treatment. Patients censored at last follow-up are indicated by a cross. The number of patients in each group at each timepoint are indicated in the table.”

We have also performed a test for interaction between rs736456 and rs3778076 and found no evidence of significant interaction. We have added some text to clarify this point to the Results section, as follows:

Results:

“There was no evidence of significant interaction between rs736456 and rs3778076 (*P* = 0.131), suggesting that each locus has an independent effect on CLL progression.”

Reviewer 1 comment 11: Figure 1 should be moved to supplemental.

Author response: Genome-wide association studies published in Nature Communications typically include the manhattan plot as a main figure. Given this, we have not moved this figure to the supplementary materials, but would be happy to do so if appropriate.

Reviewer 1 comment 12: The results are interesting and the authors show consistency across the studies. The authors conclude that the novel SNPs improve prognostication especially among those with good prognostic markers. But the authors did not jointly look among all the prognostic markers. Only one at a time. Given that these prognostic markers are typically performed in clinical practice, it is important to show that these SNPs provide information above and beyond what is already currently obtained in practice. E.g, using CLL-IPI low risk or stratified using only cases that are mutated *IGHV*, Binet stage A, and CD38 negative.

Author response: In response to the reviewers suggestion, we have included an analysis restricted to patients with low risk disease, defined as Binet stage A, *IGHV* mutated and CD38 negative (Supplementary Figure 17), demonstrating that rs736456 and rs3778076 are particularly powerful markers in this group of patients.

Specifically, we have added new text to the results as follows:

“As such, rs736456 and rs3778076 are particularly powerful prognostic markers in analysis restricted to patients with otherwise low-risk disease who have Binet stage A, *IGHV* mutated and CD38 negative disease (**Supplementary Figure 17**).”

We have also expanded our analysis to include univariate analysis of β 2 microglobulin (Supplementary Figure 4) and *TP53* status (Supplementary Figure 5), both of which are included in the CLL IPI along with *IGHV*, age and Binet stage. Unfortunately, the CLL IPI is rarely used in the clinical management of early stage CLL in the UK. As such, a complete set of CLL IPI data is available for a very small number of cases, limiting the statistical power of multivariate models.

Rather, most centres restrict the use of some IPI markers to progressive cases in need of treatment to identify those with a low probability of response to chemoimmunotherapy. Nevertheless, we have collated available data on CLL IPI markers and include new Supplementary Figures showing the prognostic power of rs736456 and rs3778076 by Binet stage, *IGHV* status, β 2 microglobulin status and *TP53* status, in addition to CD38 status (Supplementary Figures 12-16).

We have added new text to the Results section that presents these data, as follows:

“Moreover, rs736456 and rs3778076 both significantly associate with progressive disease in patients with otherwise low risk clinical markers, including *IGHV* mutation (**Supplementary Figure 13**), CD38 negativity (**Supplementary Figure 14**), low β 2 microglobulin (**Supplementary Figure 15**) and wild-type for *TP53* (**Supplementary Figure 16**). As such, rs736456 and rs3778076 are particularly powerful prognostic markers in analysis restricted to patients with disease that is Binet stage A, *IGHV* mutated and CD38 negative (**Supplementary Figure 17**). Taken together, these data identify rs736456 and rs3778076 as prognostic for disease progression in patients with otherwise low risk markers, but which have limited prognostic power in patients with high-risk markers or late stage disease.”

Reviewer #2 (Remarks to the Author):

Reviewer 2 comment 1: This is an interesting paper reporting a GWAS meta-analysis focused on time to first therapy in CLL patients. Perhaps the most interesting feature is that the SNPs seem most predictive among lower risk patients, i.e. those with Binet A disease or mutated *IGHV*, in whom we need better markers.

Author response: We thank the reviewer for their positive comments. We agree that one particular strength of our study is that the genetic markers we have identified will significantly help to predict disease progression in patients with early stage disease or who have otherwise low-risk markers.

Reviewer 2 comment 2: I have several concerns: (1) ancestry variation among the patients and cohorts, particularly given 2 of the cohorts seem to separate from the others in the PCA;

Author response: The symbols on the PCA plot (Supplementary Figure 7) represent individual CLL cases and 1000 genomes reference subjects. As such, the outlying CLL cases not defined as European were excluded from subsequent analysis. In order to clarify this we have included an inset showing the CLL cases and controls defined as European. We have modified the figure key and legend accordingly, as follows:

“Supplementary Figure 7: Principal component analysis (PCA) plot of ethnicity structure in CLL GWAS and 1000 genomes population panels. The first two principal components of the analysis are plotted for CLL cases recruited to this study (crosses). 1000 genomes European (EUR), East Asian (EAS) and African (AFR) individuals (circles) are plotted in green, purple and orange, respectively. CLL cases of European ancestry included in subsequent analysis (top-left corner), together with 1000G EUR, are further zoomed in the bottom-left gray box. PC1, principal component 1; PC2, principal component 2.”

Reviewer 2 comment 3: inadequate consideration of prognostic variables important in CLL, in particular FISH cytogenetics is not presented anywhere or included in univariate much less multivariable models; these data are essential and the new SNPs need to be found to be independent of higher risk FISH;

Author response: We have now included univariate analyses for Binet stage, *IGHV*, CD38, β 2 microglobulin and *TP53* status (Supplementary Figures 1-5). We have also included Supplementary Figures showing the prognostic power of rs736456 and rs3778076 by Binet stage, *IGHV*, CD38, β 2 microglobulin and *TP53* status (Supplementary Figures 12-16). Please see our response reviewer 1 comment 12 for further details.

Reviewer 2 comment 4: a lot of missing *IGHV* data which is perhaps the strongest predictor of TTFT;

Author response: Unfortunately, it is not standard practise to perform *IGHV* analysis for every CLL patient in the UK and the approach to testing varies considerably between clinical centres. Nevertheless, *IGHV* status was available on the majority of patients recruited to our study (Supplementary Table 1) and strongly predicts time to first treatment (Supplementary Figure 2). All multivariate models are restricted to those cases with a full set of data for those markers included in the model, thus ensuring the accuracy of these analyses. We have added text to the methods section and the figure legend to indicate this, as follows:

Methods:

"Multivariate Cox proportional hazard models incorporating known prognostic markers including *IGHV* and CD38 status were used to determine the impact of variants on disease progression. Cases with missing data for any of the variables were excluded from multivariate models."

Figure 4 legend:

Figure 4 - Multivariate prognostication models for time to first treatment (TTFT) in Binet stage A CLL incorporating genotype status for rs736456 and rs3778087. Multivariate Cox proportional hazard models for TTFT in Binet A CLL patients incorporating status for rs736456 (A) or rs3778076 (B). Hazard ratios (HR) and 95% confidence intervals (95% CI) for each variable were estimated after adjustment for GWAS and other variables in the model. Cases with missing data for any of the variables were excluded from multivariate models. (C) Kaplan-Meier survival plot for TTFT by number of cumulative risk alleles (0, 1, >1) for rs3778076 and rs736456 in Binet A CLL patients. TTFT is defined as the time from diagnosis of CLL to treatment or last follow-up without treatment. The number of patients in each group at each timepoint are indicated in the table."

Reviewer 2 comment 5: focus on CD38 which is never significant in multivariable models for CLL outcome when these other factors are included;

Author response: We included all patients in our original genome-wide analysis, but subsequently focussed analysis on Binet stage A disease to illustrate that rs736456 and rs3778076 are particularly powerful prognostic markers in patients with early stage disease. We also included analysis of *IGHV* and CD38 status because these markers are more commonly used in the clinical management of early stage CLL in the UK, although not always routinely or consistently, as highlighted by the reviewer. In contrast, testing for *TP53* is often restricted to patients with progressive disease. Nevertheless, we have now included additional analyses focussing specifically on those markers that constitute the CLL IPI. Please see our response to reviewer 1 comment 12 for details.

Reviewer 2 comment 6: it is not clear why the SNPs would predict TTFT only in Binet A, but not B and C patients. C patients need therapy right away, but the SNPs should retain prognostic significance in Binet B;

Author response: The evidence suggests that the impact of rs736456 and rs3778076 is particularly strong in early stage CLL, but which lessens as disease progresses due to the acquisition of strongly prognostic somatic markers which then dominate, including mutant *TP53* for example. Indeed, one of the markers we identified (rs3778076) is in linkage disequilibrium with an established CLL etiological risk variant on chromosome 6q21 (rs3800461) (r^2 0.46, D' 0.91) suggesting these variants drive initial CLL development and subsequent progression to symptomatic disease via a common mechanism operating during the early stages of disease development. As such, although rs736456 and rs3778076 retain some prognostic power in Binet stage B/C patients their impact weakens with the acquisition of strongly prognostic somatic markers. We have added some additional text to the discussion highlighting this model, as follows:

Discussion

"Our data suggest that rs736456 and rs3778076 are powerful prognostic markers in early stage CLL, but have less impact as disease progresses, predicted to be due to the acquisition of strongly prognostic somatic alterations which then dominate. Consistent with this model, the prognostic power of rs3778076 and rs735456 is weaker in patients with Binet stage B or C disease and in patients with high risk somatic markers."

Reviewer 2 comment 7: Given there is no control group, I think an independent validation cohort demonstrating that these SNPs still have independent prognostic significance for TTFT is needed.

There is a long history of findings that do not hold up in validation cohorts.

Author response: We have sourced data and samples from a seventh CLL cohort and directly genotyped these for rs736456 and rs3778076. We have updated the Forest plot shown in Figure 3 to include the new cohort and also performed new survival analyses and re-plotted all of the Kaplan-Meier curves throughout the manuscript to include the new cases. Multivariate models also include the new cases where the relevant data is available. New text has been added to the Abstract, Results and Methods sections, as follows:

Abstract:

“Prognostication in patients with chronic lymphocytic leukemia (CLL) is challenging due to a considerable heterogeneity in clinical course. We hypothesized that constitutional genetic variation affects disease progression and could aid in prognostication. Pooling data from seven studies incorporating 842 cases identified two genomic locations associated with time from diagnosis to treatment, including 10q26.13 (rs736456, hazard ratio (HR) = 1.78, 95% confidence interval (CI) = 1.47-2.15; $P = 2.71 \times 10^{-9}$) and 6p (rs3778076, HR = 1.99, 95% CI = 1.55-2.55; $P = 5.08 \times 10^{-8}$), which were particularly powerful prognostic markers in patients with early stage CLL. Expression quantitative trait loci analysis identified putative functional genes implicated in modulating B-cell receptor or innate immune responses, key pathways in CLL pathogenesis. These data identify rs736456 and rs3778076 as prognostic in CLL demonstrating that disease progression is determined by constitutional genetic variation as well as known somatic drivers.”

Results:

“We genotyped rs736456 and rs3778076 in a seventh cohort, bringing the total number of CLL cases analysed to 842. Both markers showed consistent direction and magnitude of effect sizes across all seven studies with no evidence of heterogeneity (Figure 3).”

Methods:

“Fidelity of array genotyping and imputed dosages was confirmed using Sanger sequencing in a subset of samples for each sentinel variant with concordance of 100% for all variants (Supplementary Figure 8). rs3778076 and rs736456 were directly genotyped in study 7 using Sanger sequencing.”

Reviewer 2 comment 8: the discussion of prognostic markers in the Introduction is very naïve from a clinical CLL point of view, should be better prioritized. In particular, CD38 can vary from day to day within a patient, and is not a stable prognostic marker, so I would not focus so much on it.

Author response: We have re-structured the introductory section discussing prognostic markers in CLL, as follows:

“Prognostication in early stage CLL and identifying those patients at risk of progression remains challenging. Numerous methods of CLL prognostication exist, which range from clinical assessments to genomic techniques. Clinical assessments include the Rai or Binet scores, where adverse outcomes are associated with increasing tumour load and evidence of marrow failure, and the CLL international prognostic index (CLL-IPI) which utilises a combination of clinical and biological characteristics to assess prognosis. ~~Biological prognostic tests include CLL-cell expression of antigens such as CD38, zeta-associated protein 70 (ZAP-70) and CD49d⁸ identify patients with a poor outcome, as do deletions in the long arm of chromosome 11 (del(11q)) and the short arm of chromosome 17 (del(17p))⁹ although these alterations are infrequent in newly diagnosed CLL patients^{10,14}.~~ Cases with unmutated immunoglobulin heavy chain variable region (IGHV) genes¹² and those with mutations/deletions affecting TP53, NOTCH1 and SF3B1 have poor prognoses^{13,14}, although these alterations are infrequent in newly diagnosed CLL patients^{10,11}. Once patients progress to symptomatic disease then somatic drivers of disease, particularly mutations of TP53 and SF3B1 as well as complex karyotype, become dominant prognostic markers^{15,16}. ~~Post-therapy minimal residual disease levels have emerged as key indicators of post-therapy prognosis and are probably indirect markers of the chemosensitivity of the underlying CLL clone to the therapy administered¹⁷.~~”

Reviewer 2 comment 9: supplementary figure 8 – the colors for the ancestries are not visible and

neither are all the GWAS visible. Looks like GWAS 1 and 2 are separated from the others. excluding non-Europeans based on PCA analysis using 1000 genome data is not really adequate control for ancestry variation, particularly as the PCA suggests some ancestry heterogeneity.

Author response: Please see author response to reviewer 2 comment 2. In order to make the individuals more visible we have included an inset showing the CLL cases and controls defined as European.

Reviewer 2 comment 10: similarly the comment that CD38 positive disease along with unmutated IGHV and Binet B or C disease predict shorter TTFT and OS in univariate analysis is inadequate to really address this cohort. We need a full report of all factors associated with TTFT and OS in univariate analysis, as well as multivariate analysis.

Author response: In addition to univariate analyses for *IGHV*, Binet stage and CD38 status, we have included new univariate analyses for $\beta 2$ microglobulin and *TP53* status (Supplementary Figures 1-5). We have included new Supplementary Figures showing the prognostic power of rs736456 and rs3778076 in combination with each of the CLL IPI markers, including Binet stage, *IGHV* status, $\beta 2$ microglobulin status and *TP53* status, in addition to CD38 status (Supplementary Figures 12-16). Please see the author response to reviewer 1 comment 12 for details.

Reviewer 2 comment 11: It furthermore appears that there are a LOT of missing data on IGHV, which has to be very well controlled for in this analysis. How was missing IGHV data handled in the multivariable analysis?

Author response: Cases with missing data for any of the variables were excluded from multivariate models. Please see the author response to reviewer 2 comment 4.

Reviewer 2 comment 12: 5 variants look separated on the QQ plot but only 2 come up in the analysis? What are the others and how do they look?

Author response: The reviewer is correct in that a total of 5 SNPs reached genome-wide significance in meta-analysis, including two additional SNPs at 10q26.13 (rs4752676 and rs736457) and one additional SNP (rs11757517) at 6p. These three additional SNPs are in high linkage disequilibrium ($r^2 > 0.8$) with the corresponding index SNPs that have the smallest P values at 10q26 (rs736456) and 6p (rs3778076). After conditional analysis on each index SNP there is little evidence of secondary signals ($p < 10^{-4}$). Thus, the 2 index SNPs were used in the subsequent analysis. For completeness, we have included new regional and forest plots for these 3 additional SNPs in Supplementary Figures 60-62 and have added some additional text to the legend of Figure 2 to highlight these variants, as follows:

“Two additional SNPs at 10q26.13 (rs4752676 and rs736457) (A) and one additional SNP at 6p (rs11757517) (B) also reach genome-wide significance in meta-analysis (Supplementary Figures 60-61). The symbol for rs11757517 is obscured on the regional plot (B) by the symbol for rs3778076.”

Reviewer 2 comment 13: the Discussion states the CLL is a precursor B cell neoplasm; this is incorrect, it is a mature B cell neoplasm. In addition the first half of the Discussion is mostly pure speculation about CLL causation that has nothing to do with this paper and should be modified.

Author response: We have removed the first two paragraphs of the Discussion and now restrict discussion to the results presented in our manuscript, as follows:

“DISCUSSION

~~CLL is thought to arise from a B-cell progenitor which develops under the influence of chronic antigenic stimulation and micro-environmental influences into a precursor B-cell neoplasm. This multi-step process may be initiated in a haematopoietic stem cell with the consequent production of abnormal pro-B lymphocytes and expanded populations of CD5+ lymphocytes, postulated precursors of CLL. Abnormal antigen mediated selection of a subset of either CD5+ lymphocytes or marginal~~

~~zone-derived lymphocytes associated with a breakdown in normal immunological tolerance may permit the expansion of a clonal CLL population. The antigenic drive for this clonal expansion is unknown but could represent a response to autoantigens, viral proteins or autonomous antigen-independent activation of the BCR.~~

~~The clinical course and prognosis of CLL can be related to the possible cell of origin by analysis of the extent of *IGHV* mutation, with a poor prognosis linked to the presence of relatively unmutated *IGHV* genes, possibly suggesting a pre-germinal cell of origin or a non-germinal centre dependent maturation process. The acquisition of somatic driver mutations such as those occurring in *TP53*, *Notch1*, *ATM* and *SF3B1*, or abnormal patterns of somatic gene methylation also contribute to disease progression.~~

CLL susceptibility loci identified to date implicate genes central to B-cell development, BCR signalling and apoptotic responses. These risk alleles may therefore function by promoting the generation of abnormal pro-B populations or to maintain the outgrowth of CLL-precursors in response to continuous BCR signalling. However, the current model would suggest that once the pre-leukemic CLL clone has established disease progression becomes wholly dependent on the classic multi-hit model of oncogenesis where acquired somatic drivers promote clonal expansion.”

We would like to thank the reviewers for their insightful comments.

REVIEWER COMMENTS

Reviewer #1 (Remarks to the Author):

The authors responded well to my earlier comments. I have no further comments.

Reviewer #3, Replacement Reviewer for Reviewer #2 (Remarks to the Author):

This manuscript presents a study of germline genetic variants in CLL associated with time to first treatment and identifies two genomic risk loci associated with shorter time to first treatment in patients with lower risk CLL. There are several limitations with this manuscript:

1) While the authors include many common clinically utilized prognostic variables here (Binet Stage, IGHV mutational status, TP53 mutations/deletions, etc), there are other potential prognostic factors not assessed (eg NOTCH1 & SF3B1 mutations). Moreover, identifying a prognosticator limited to Binet Stage A disease is a somewhat limited advance.

a. As one minor point related to the above, the sentence in the Introduction stating "those with mutations/deletions affecting TP53, NOTCH1, and SF3B1 have poor prognosis" is incorrect. NOTCH1 and SF3B1 are not affected by deletions in CLL, but by somatic mutations (whereas TP53 can be affected by mutations and/or deletions). This sentence needs to be revised.

2) The inference of cis regulated genes associated with the 2 risk loci was performed using gene expression data from blood of normal subjects. While clearly such a large collection of gene expression data for CLL samples is not available, it seems quite possible that gene expression data from CLL patients may yield gene expression results that are distinct from those in normal blood samples. Thus, the data on the association of these risk alleles with altered expression of specific genes does not appear strong.

In addition to the above issues, it is unclear to me if the 6 GWAS studies cited have previously been published or if these data are being published for the first time. It is also not clear to me whether/where these genetic data are deposited. I only see mention of summary statistics and

Reviewer #4, Replacement Reviewer for Reviewer #2 (Remarks to the Author):

The authors have addressed nearly all of the queries adequately.

this reviewer would like to see a little more nuance and care in reporting the main finding- the two SNPs are most informative in early stage, low risk CLL patients.

a point that is not fully discussed is the temporal nature of SNPs- which do not epistatically work in parallel- and here is a place where a subset could be informative in select circumstance. It is a shame not to have more data on the evaluation early on but it is understandable.

There is one query that needs further detailed attention:

The limited sample size with apparent ancestry heterogeneity (PCA in Suppl Fig 7) could be a critical issue. Although they excluded non-European individuals, this does not seem to be sufficient to deal with the apparent two subgroups. A PCA with more reference populations would be interesting to check if this is due to a limited diversity in the reference populations used, or due to an ethnicity structure that could potentially limit the interpretation of the results.

Reviewer #4 (replacement for Reviewer 2) (Remarks to the Author):

Reviewer 4 comment 1: The authors have addressed nearly all of the queries adequately. this reviewer would like to see a little more nuance and care in reporting the main finding- the two snps are most informative in early stage, low risk CLL patients. a point that is not fully discussed is the temporal nature fo SNPs- which do not epistatically work in parallel- and here is a place where a subset could be informative in select circumstance. It is a shame not to have more data on the evaluation early on but it is understandable.

Author response: In response to the reviewers request for a more nuanced reporting of the main finding (that the two SNPs are most informative in early stage, low risk CLL) we have included additional text to the abstract and discussion, as follows:

Abstract

“Pooling data from seven studies incorporating 842 cases identified two genomic locations associated with time from diagnosis to treatment, including 10q26.13 (rs736456, hazard ratio (HR) = 1.78, 95% confidence interval (CI) = 1.47-2.15; $P = 2.71 \times 10^{-9}$) and 6p (rs3778076, HR = 1.99, 95% CI = 1.55-2.55; $P = 5.08 \times 10^{-8}$), which were particularly powerful prognostic markers in patients with early stage CLL **otherwise characterised by low-risk features.**”

Discussion

“The evidence presented herein suggests that, in addition to being etiological, constitutional risk alleles contribute to disease progression in CLL post-diagnosis. Our data suggest that rs736456 and rs3778076 are powerful prognostic markers in early stage CLL, but have less impact as disease progresses, predicted to be due to the acquisition of strongly prognostic somatic alterations which then dominate. Consistent with this model, the prognostic power of rs3778076 and rs735456 is weaker in patients with Binet stage B or C disease and in patients with high risk somatic markers, **such as unmutated IGHV or CD38 positivity.**”

And also here in the Discusson:

“Genetic tests for constitutional genetic markers, such as rs736456 and rs3778076, have the advantage of being easy to perform, highly reproducible and inexpensive making them ideal for incorporation into multivariate prognostication models for disease progression in **early stage CLL.**”

The reviewers point on temporal nature of the SNPs is pertinent and we have added a comment to the discussion to highlight this aspect, as follows:

Discussion

“CLL susceptibility loci identified to date implicate genes central to B-cell development, BCR signalling and apoptotic responses²⁰. These risk alleles may therefore function by promoting the generation of abnormal pro-B populations or to maintain the outgrowth of CLL-precursors in response to continuous BCR signalling. However, the current model would suggest that once the pre-leukemic CLL clone has established, disease progression becomes **largely** dependent on the classic multi-hit model of oncogenesis where acquired somatic drivers promote clonal expansion¹¹, **thus temporally limiting the impact of constitutional genetic variants to disease progression at an early stage.**”

Reviewer 4 comment 2: There is one query that needs further detailed attention: The limited sample size with apparent ancestry heterogeneity (PCA in Suppl Fig 7) could be a critical issue. Although they excluded non-European individuals, this does not seem to be sufficient to deal with the apparent two subgroups. A PCA with more reference populations would be interesting to check if this is due to a limited diversity in the reference populations used, or due to an ethnicity structure that could potentially limit the interpretation of the results.

Author response: To the best of our knowledge the European panel from the 1000 Genomes (1000G) project comprising CEU (Utah Residents (CEPH) with Northern and Western European Ancestry), GBR (British in England and Scotland), FIN (Finnish in Finland), IBS (Iberian Population in Spain) and TSI (Toscani in Italia) is the most diverse publicly available dataset.

In order to address the reviewers query about ancestral heterogeneity we have performed some additional analysis. Principal component analysis (PCA) of the 1000G European sub-panel and CLL cases shows a small number of cases with principal component (PC) 1 > -0.013, but these are still within the European PC1 boundary defined by the European population (Figure 1, top left and inset). We subsequently performed a European case-only PCA which identified 13 CLL cases with PC1 > 0.1, distributed across 4 of the 6 GWAS cohorts (Figure 2). In order to be test whether these cases were driving the reported associations between the two SNPs and disease progression we repeated the survival analysis with these 13 cases excluded, which confirmed genome-wide significant associations for rs3778076 ($p=1.73 \times 10^{-8}$) and rs735456 ($p=7.89 \times 10^{-9}$) with time to first treatment (Figures 3 and 4). As such, we are confident that any ancestral heterogeneity in the CLL cases is not responsible for driving the reported associations. Given that these 13 cases are defined as European by current GWAS criteria (and to avoid any criticism of applying artificial selection criteria) we would prefer to retain them in the analysis, but they can be removed if required. Nevertheless, in order to aid interpretation of the data we have modified Supplementary Figure 7 to show the different sub-populations that make-up the European panel in the 1000G project (see Figure 1 below).

We would like to thank the reviewers for their insightful comments.

Figures

Figure 1 - Principal component analysis (PCA) plot of ethnicity structure in CLL GWAS and 1000 genomes population panels. The first two principal components of the analysis are plotted for CLL cases recruited to this study (crosses). 1000 genomes European (EUR), East Asian (EAS) and African (AFR) individuals (open circles) are plotted in brown, pink and gray, respectively. CLL cases of European ancestry included in subsequent analysis (top-left corner) are shown in the inset together with 1000G EUR sub-populations (closed circles) comprising CEU (Utah Residents (CEPH) with Northern and Western European Ancestry; pink), GBR (British in England and Scotland; dark green), FIN (Finnish in Finland; orange), IBS (Iberian Population in Spain; blue), and TSI (Toscani in Italia; light green). PC1, principal component 1; PC2, principal component 2.

Figure 2 - Principal component analysis (PCA) plot of ethnicity structure in CLL GWAS cases. The first two principal components of the analysis are plotted for CLL cases recruited to GWAS1, GWAS2, GWAS3, GWAS4, GWAS5 and GWAS6. PC1, principal component 1; PC2, principal component 2.

Figure 3 – Forest plot showing association between rs3778076 and time to first treatment excluding 13 CLL cases with PC1 > 0.1. Forest plot for rs3778076 and age-adjusted association with time to first treatment (TTFT). TTFT is defined as the time from diagnosis of CLL to treatment or last follow-up without treatment. No/events: Number of CLL patients/Number of patients receiving treatment; Eff/Ref: effect/reference allele; EAF: effect allele frequency; HR: hazard ratio; CI: confidence interval; Squares denote the per-allele HR, with size proportional to the weight of the study. Pooled HRs derived from both the fixed and random-effects models are indicated by diamonds with their corresponding meta P values shown in the left parentheses. X-axis label formats include reference sequence (rs) identifier and chromosome:position (b37). P values for Cochran's Q test (P_{het}) and I² for heterogeneity are shown in parentheses.

Figure 4 – Forest plot showing association between rs736456 and time to first treatment excluding 13 CLL cases with PC1 > 0.1. Forest plot for rs736456 and age-adjusted association with time to first treatment (TTFT). TTFT is defined as the time from diagnosis of CLL to treatment or last follow-up without treatment. No/events: Number of CLL patients/Number of patients receiving treatment; Eff/Ref: effect/reference allele; EAF: effect allele frequency; HR: hazard ratio; CI: confidence interval; Squares denote the per-allele HR, with size proportional to the weight of the study. Pooled HRs derived from both the fixed and random-effects models are indicated by diamonds with their corresponding meta P values shown in the left parentheses. X-axis label formats include reference sequence (rs) identifier and chromosome:position (b37). P values for Cochran's Q test (P_{het}) and I² for heterogeneity are shown in parentheses.

REVIEWERS' COMMENTS

Reviewer #4 (Remarks to the Author):

Thank you for the thoughtful response, particularly addressing the PCA. This reviewer is fully satisfied with the responses